# Investigation of the Frying Fume Composition During Deep Frying of Tempeh Using GC-MS and PTR-MS

**DOI:** 10.3390/molecules29215046

**Published:** 2024-10-25

**Authors:** Rohmah Nur Fathimah, Tomasz Majchrzak

**Affiliations:** Department of Analytical Chemistry, Faculty of Chemistry, Gdańsk University of Technology, 80-233 Gdańsk, Poland; rohmahfathimah@gmail.com

**Keywords:** deep frying, tempeh, PTR-MS, GC-MS, food processing, VOCs

## Abstract

This study employed proton transfer reaction mass spectrometry (PTR-MS) and gas chromatography–mass spectrometry (GC-MS) to identify and monitor volatile organic compounds (VOCs) in frying fumes generated during the deep frying of tempeh. The research aimed to assess the impact of frying conditions, including frying temperature, oil type, and repeated use cycles, on the formation of thermal decomposition products. A total of 78 VOCs were identified, with 42 common to both rapeseed and palm oil. An algorithm based on cosine similarity was proposed to group variables, resulting in six distinct emission clusters. The findings highlighted the prominence of saturated and unsaturated aldehydes, underscoring the role of fatty acid oxidation in shaping the frying fume composition. This study not only corroborates previous research but also provides new insights into VOC emissions during deep frying, particularly regarding the specific emission profiles of certain compound groups and the influence of frying conditions on these profiles.

## 1. Introduction

Deep frying is one of the most widely applied cooking methods because it imparts a desirable appearance, texture, and flavour to the finished product. This technique involves immersing food in heated oil at temperatures ranging from 150 to 200 °C [1,2,3], typically until achieving a golden brown colour. During frying, a plethora of VOCs are formed such as aldehydes, esters, acids, aromatics, ketones, pyrazines, furans, and alcohols [4,5]. These compounds are the result of the intricate chemical transformations triggered by heat, including hydrolysis, oxidation, and polymerisation [6], which play a significant role in producing, stabilising, and affecting the quality of fried food, frying oil, and the produced volatile compounds [7,8,9]. Altering parameters such as the temperature, duration, oil’s fatty acid composition, and fried food’s properties can lead to marked variations in the resulting volatile compounds [10].

Tempeh is a rich source of protein originating in Indonesia, made from soybean fermented by the fungus *Rhizopus oligosporus*. It remains the most favoured soy-based food and an essential component of the Indonesian diet for its affordability and nutritional value [11]. Typically, tempeh is deep-fried before consumption or further cooked, often perceived as a meat substitute. Besides serving as a daily meal, tempeh is deeply rooted in Indonesian culture, particularly during festive occasions, religious ceremonies, and local traditions. For example, in rural areas of Central Java and Yogyakarta, tempeh symbolises gratitude and respect for the cycle of life in local spiritual beliefs when offered during the *kenduri* or *selamatan* ceremonies [12]. The growing interest has moved people’s perception of tempeh from a low-class food to a widely accepted and affordable meat alternative consumed by people across socio-economic backgrounds [13]. It is mostly associated with its potential health benefits, as tempeh is rich in bioactive compounds, prebiotics, and a wide array of vitamins, contributing to its antioxidant and antimicrobial properties [14,15,16].

The widely applied method for identifying volatile compounds is gas chromatography–mass spectrometry (GC-MS). This technique can be applied for targeted analysis, such as PAH emission during frying [17], VOCs, such as aldehydes, ketones, and fatty acids, and BTEX, among others [18]. In recent years, experiments have been oriented towards holistic, untargeted analysis, including two-dimensional gas chromatography–mass spectrometry (GC × GC-MS) [19]. While GC-MS remains an excellent technique, its limitations become increasingly apparent when tracing chemical compositions in real time. Achieving a comprehensive understanding of the emitted volatile compounds, characterised by intricate patterns, demands approaches that transcend the bounds of compound-specific identification. Innovative technologies such as proton transfer reaction mass spectrometry (PTR-MS) have emerged in response to this need.

PTR-MS offers rapid and real-time monitoring of VOCs with high sensitivity, eliminating the need for labour-intensive sample preparation processes [20]. It also possesses high sensitivity and a low limit of detection (LOD) down to parts per trillion by volume (ppt_v_) [21]. PTR-MS is widely applied in food analysis, including food shelf life [22], origin [23], and taste and flavour studies [24]. Due to its ability for real-time measurement, this technique finds its application in the monitoring of food processing, including roasting, cooking, baking, or frying [25]. Nevertheless, these merits come with an inherent limitation—the identification of compounds can be challenging. The absence of a specific separation step within PTR-MS and its reliance on distinguishing compounds solely through their mass-to-charge (*m*/*z*) values often underestimate the volatile composition. Thus, combining PTR-MS with GC-MS for enhanced accuracy and compound identification is advisable.

The complementary use of GC-MS and PTR-MS is still not a widespread solution, but examples of their joint use in medical, environmental, and also food research can be found in the literature [20]. It may be noteworthy that the research on the key aroma biomarkers of blueberries distinguishes groups based on their genetic background and maturity stages [26], and the species-specific characterisation of white truffles [27]. The combination of the two techniques has also been applied to investigate the dynamic shifts in aroma release and flavour perception during the drinking of alcoholic beverages [28], and in the potato frying process, showing insights into early-stage VOC release and oil degradation [29].

Unlike GC-MS, PTR-MS generates data as counts per second, offering a high temporal resolution. It allows for the comprehensive exploration of the generated VOC through innovative data analysis approaches. Interestingly, the information on VOC emissions, including their fluctuations and underlying patterns, provides valuable insights into the field of foodomics. Furthermore, applying advanced multivariate statistics, often called chemometrics, proves invaluable for this type of study. Through this approach, a more nuanced exploration of the intricate relationship between VOCs and the frying process can be carried out by understanding the alteration pattern of these emissions.

Thus, this study aims to uncover distinctions arising from different tempeh frying conditions and discern patterns of compound emissions and similarities across various frying scenarios. Aside from the different temperatures and types of vegetable oil used, the study introduced the element of repeated frying cycles. Frying was subjected to five consecutive frying cycles, simulating continuous frying processes. A multivariate analysis of hierarchical clustering was chosen for the study’s data exploration. By employing this method, valuable insights into the potential similarities or dissimilarities in volatile compound emissions across different frying conditions can be extracted. Dynamic perspectives on the emitted VOCs over time are also presented, offering deeper insights compared to single-stage or static processes.

## 2. Results and Discussion

### 2.1. Volatile Component Profiling of Used Oils via HS-SPME-GC-MS

To gain insight into the volatile organic compounds produced during tempeh deep frying, untargeted analyses were performed via HS-SPME-GC-MS for both palm oil (PO) and rapeseed oil (RO). These oil samples were obtained after frying tempeh at 180 °C. A total of 78 volatile organic components were identified, encompassing 4 acids, 12 alcohols, 21 aldehydes, 19 alkanes, 2 benzene derivatives, 9 furans, 9 ketones, and 2 other compounds (see Figure 1A).

The volatile components displayed divergence between palm and rapeseed oil with RO containing 58 of these compounds, while PO exhibited a slightly broader volatile profile with 62 compounds. Interestingly, 42 volatile compounds were shared between the two oils. However, the disparity between the compounds is more pronounced, with 16 of them exclusive to RO and a notably higher count of 20 unique to PO.

An in-depth analysis of the shared VOCs is provided in Figure 1B. It is possible to indicate not only that palm oil possesses more compounds but also comprises higher-intensity volatile compounds than rapeseed oil across all groups of compounds. The compounds that possess the higher intensity in palm oil are mostly saturated aldehydes, such as hexanal, heptanal, and nonenal. Rapeseed oil is richer in alkenals and alkadienals, such as 2-propenal, 2-butenal, and 2,4-heptadienal. These aldehydes are the oxidation products of unsaturated fatty acids. For example, nonanal is the oxidation product of oleic acid, hexanal of linoleic acid, and 2,4-heptadienal of α-linolenic acid [30].

The headspace experiment therefore made it possible to identify key oil degradation products during tempeh frying and to determine differences in the profiles of emitted VOCs. This is a valuable insight for the next step, namely measuring the composition of oil fumes during frying.

### 2.2. Emission of Volatile Compound Using PTR-MS

In each experimental trial, the chosen oils were progressively heated. To guarantee stabilisation, the oil was kept at the desired temperature for sixty seconds after the intended frying temperatures of 160 °C and 180 °C were reached to create a solid baseline for the frying process and the subsequent data analysis. After the temperature reached a stable point, tempeh was added to the oil while keeping the system closed to capture all fumes. To fully observe the extended frying phase, the tempeh was fried for three minutes, or until it turned a dark brown or burnt colour.

An in-house R script was then introduced to process all of the data files from the different frying conditions. The main purpose was to group ion signals based on their emission patterns regardless of factors like temperature, time, or type of oils. This means examining if the released compounds behaved similarly in different experimental settings by concentrating on lipid degradation and volatile compound emission patterns. A big dataset was prepared by combining the preprocessed data from all experimental runs, which included 12 different frying conditions with three replications for each setting, resulting in a dataset comprising a total of 208,512 data points. The final data matrix was further analysed by computing the cosine similarities and used to construct a cluster analysis (CA).

A depiction of the CS between registered ions is presented in Figure 2. It was possible to determine six dominant clusters, which suggest six distinctive emission patterns. The reason for the shared emission profile for different ions could lie in two factors. Firstly, the ions can result from the compounds that follow the same chemistry, e.g., the same chemical pathway of formation, and the same substrate (same fatty acid, peroxides, etc.), or they could be formed in the drift tube of the PTR-MS instrument due to the fragmentation. Emission patterns for the individual ions in the different frying conditions are shown in the Appendix A. The identification of the possible compounds is presented in Table 1 together with the retention time from the HS-SPME-GC-MS experiment.

Thus, in Cluster 1, the two ions shared a similar emission pattern. These ions—153.13 *m/z* and 155.14 *m/z*—are identified as C_10_-unsaturated aldehydes, namely 2,4-decadienal and 2-decenal. Additionally, both compounds originate from the same primary oxidation compound, 9-hydroperoxide, with 2,4-decadienal recognised as the decomposition product and 2-decenal as the product of beta haemolysis [31]. The observed behaviour of these compounds during frying can be attributed to the delicate balance between their production and consumption rates, which resulted in a linear emission pattern. The strong fluctuations of the signal could result in low-intensity signals and close-to-noise readings.

In Cluster 2, the steep emission curve is obtained in the first minute of frying, and the near-plateau signals were registered later. An exception was the frying at 160 °C, where 2-propenal rapidly increased after 20 s of frying. This cluster is dominated by the alkenals, namely 2-propenal (acrolein), which has the most intensive signal, 2-butenal, 2-methyl-2-propenal (methacrolein), and 2-hexenal. The high level of acrolein in the frying fumes is a result of its volatility as well as the formation pathways, both by (1) the hydrolysis of the triglycerides followed by free radical reactions and (2) the peroxidation of polyunsaturated fatty acids (PUFAs) [32]. The peroxidation of PUFAs is indeed one of the key factors in the formation of saturated and unsaturated aldehydes [33]. In this cluster, 2,4-heptadienal was also detected. Additionally, Cluster 2 could be divided into two sub-clusters. The first contains the C3–C5 aldehydes, and the second contains 2-hexenal and 2,4-hexadienal together with its ion fragments, namely 67.05 *m/z* and 81.07 *m/z*. The unsaturated aldehydes are the key thermal degradation product of various fatty acids and were monitored in previous studies [34]. Their high reactivity and toxicity are often addressed, especially for acrolein [35]. A study shows that workers in the food industry, especially cooks and chefs, are at serious risk for occupational health problems from the acrolein emitted during the frying process, particularly in poorly ventilated environments [36]. Carcinogenic and non-carcinogenic risks are linked to acrolein exposure, primarily leading to respiratory injuries and other long-term health problems [37,38].

Cluster 3 is characterised by a rapid increase in emissions in the first few seconds of frying in most of the frying conditions. The identified compounds that are assigned to this cluster are furan (69.03 *m*/*z*), butanal (73.06 *m*/*z*; found in palm oil only), 2-methyl-tetrahydrofuran, and pentanal (87.08 *m*/*z*). Furan is a product of 4-hydroxy-2-alkenal cyclisation [39]. The possible tracing of 2-methyl-pentane (87.12 *m*/*z*), which was identified during the HS-SPME-GC-MS experiment, is unlikely since its proton affinity value (PA; for pentane and hexane, it is in the range of 721 ± 20 kJ mol^−1^ [40]) could be close to water’s PA (approx. 690 kJ mol^−1^ [41]). The ion signal of 97.06 *m*/*z* could be assigned as 2-ethylfuran; however, it was not confirmed during the HS-SPME-GC-MS experiment. This compound was found during perilla oil thermal oxidation [42] and can be formed from the 2-hexenal through cyclisation [43]. The ions 41.04 *m*/*z* and 43.05 *m*/*z* are common ion fragments, and it is hard to assign them to particular compounds. To conclude, Cluster 3 consists mainly of short-chained unsaturated aldehydes and furans.

There are only two compounds clustered in Cluster 4, namely ethanol (47.05 *m*/*z*) and acetaldehyde (45.03 *m*/*z*). Acetaldehyde is one of the key products of food thermal processing and was reported in various experiments [18,44,45,46,47]. It can be formed in lipid peroxidation together with other aldehydes, as referenced previously. There are a limited number of studies that found ethanol in frying fumes, and it is speculated that it can be formed during lipid degradation [48].

The three ions that form the isolated Cluster 5 are 55.05, 83.09 *m*/*z*, and 101.10 *m*/*z*. Hexanal, identified through 101.10 *m*/*z*, exhibits a fragmentation pattern that generates fragment ions at *m*/*z* 83.09 (C_6_H_11_^+^) and 55.05 (C_4_H_7_^+^) (verified experimentally; see Appendix A in Appendix A). The ion 55.05 *m*/*z* signal can result from the appearance of water cluster H_7_O_3_^+^ (55.04 *m*/*z*) and can be considered an additive signal for both ions. Hexanal can be formed during the oxidation of n-6 PUFAs [49] and was determined and monitored in our previous experiments [29].

The biggest cluster, Cluster 6, consists of 11 ion signals and grouped the signals characterised by the close-to-linear emission for the low cycle of frying (frying temperature of 180 °C—first and second frying—and at 160 °C). For the rest, frying scenarios exhibit a flattened emission curve after one minute of frying. This cluster is represented by the ions with relatively high *m*/*z* values, ranging from 95.05 *m*/*z* to 143.18 *m*/*z*. Most of them are isobaric; thus, it is impossible to distinguish their particular emission characteristics. The list of the identified compounds consists of C_8_–C_10_ alkanes (nine compounds), C_7_–C_9_-saturated (three) and -unsaturated (three) aldehydes, alcohols (two), ketones (two), 1-octene, and 2-pentyl furan. The 2-pentyl furan can be formed during the cyclisation of 2-nonenal [37]; thus, their emission profiles are correlated (R_Pearson_ = 0.879).

### 2.3. Impact of Repeated Use Cycles, Temperature, and Type of Oil on VOC Emission Profile

Repeated use cycles, where the same batch of oil is reused multiple times for frying, is a common practice. Due to various chemical changes, primarily lipid oxidation and hydrolysis caused by exposure to atmospheric oxygen, high temperatures, and interactions with foodstuffs (e.g., moisture content) [50], the quality of the oil gradually degrades [51]. These degradation processes result in the release of VOCs into the frying fumes. For two types of oil, rapeseed and palm oil, and across five cycles of repeated frying, the emission patterns of clustered volatiles are presented in Figure 3 (rapeseed oil) and Figure 4 (palm oil). The individual emission plots are shown in the Appendix A.

Regardless of the oil used, the profile of monitored compounds remained consistent within each cluster. Cluster 1 (C_10_-unsaturated aldehydes) exhibited the lowest emissions compared to other clusters, with a nearly linear emission trend. However, the true emission pattern is hard to explore, mainly due to the close-to-baseline signal, especially for fresh oil. For rapeseed oil experiments only, the slope of the emission curve was slightly affected by the immersion of tempeh, indicating that the emission of 2-decenal and 2,4-decadienal can be accelerated while food is placed. In palm oil, on the contrary, the effect of frying was less pronounced, except for the fourth and fifth frying cycles, where close-to-exponential growth was observed. In all frying scenarios, the emission intensity gradually increases across use cycles, resulting in a steep emission curve during the fifth reuse cycle.

Cluster 2 displayed a near-logarithmic growth, with 2-propenal (acrolein) as the most intense compound (see Appendix A). There, the tendency to form a plateau after approx. one minute of frying was observed for most of the registered ions. Acrolein was the dominant VOC emitted in the majority of frying scenarios, except frying in fresh rapeseed oil where propanal was the highest. The second highest signal was 71 *m*/*z* (methacrolein and 2-butenal), which exhibits a similar emission pattern to the acrolein. Interestingly, 71 *m*/*z* had a similar emission intensity as acrolein in the scenarios where rapeseed oil was used and the frying temperature was set as 180 °C. In the lower temperature and all palm oil scenarios, the emission of acrolein dominates. Moreover, in runs where rapeseed oil was used and 160 °C was set as the temperature, the emission rapidly grew when immersing tempeh and did not form a plateau. In the case of rapeseed oil, emissions remained consistent across use cycles, while for palm oil, emissions increased after the third cycle. Furthermore, emissions from rapeseed oil were approximately 2–3 times higher than those from palm oil, likely due to differences in fatty acid composition and oxidative stability [52]. In most cases, the strong influence of the immersion of tempeh was observed.

In Cluster 3, food immersion had a significant impact, with emissions sharply increasing in the first 20 s of frying, followed by a behaviour similar to Cluster 2 compounds, but less intense. Furan was the most emitted volatile in this cluster in the scenarios where palm oil was used. As furan can originate from both oil [53] and oil–food interactions [54,55], the rapid increase during the initial frying phase is justified. Contrary to palm oil, frying with rapeseed oil promotes ethyl furan over furan emission. In those scenarios, the very steep emission of ethyl furan can be registered. It was possible to capture a bursting peak while fresh rapeseed oil was used while frying at 180 °C. In our previous research, we speculated that the reason for this peak-like signal rise might be due to the water bubbles that escape the oil bulk in the first seconds of frying [34]. Unlike Cluster 2, the emission of Cluster 3 volatiles increased after the third cycle when rapeseed oil was used, while it remained stable with palm oil.

Cluster 4, comprising acetaldehyde and ethanol, showed a similar emission pattern to Cluster 2, with a flattening of emissions after approximately 100 s and a slight decrease in the later stages of frying. The strong influence of the water within the food can be postulated since a bursting peak in the first seconds was registered for some experiments. In palm oil, emissions increased from the fourth frying cycle. All experiments exhibited strong signal fluctuations, likely due to the high volatility of these compounds causing losses at connections and seals. The emissions of Cluster 4 VOCs were 2–3 times higher in rapeseed oil than in palm oil. For both oils, where 180 °C was applied for frying, acetaldehyde and ethanol exhibit similar intensity. However, in the case of 160 °C, the ethanol emission was higher in comparison to acetaldehyde.

Hexanal (Cluster 5) exhibited a similar emission pattern to the Cluster 2 compounds, with a rapid increase observed in the first 5 s of frying. Ion fragments were much higher than MH^+^ ions and displayed comparable intensity. Hexanal emissions were higher in palm oil, particularly during the first frying cycle.

Lastly, Cluster 6 volatiles followed a near-linear emission pattern, with overall intensities higher than those of Cluster 1. However, during the first frying cycle, the emissions resembled those of Cluster 2. A clear increase in emissions over repeated cycles was observed, beginning after the second cycle in rapeseed oil and after the third cycle in palm oil. The most dominant for both oils were two ions, 95 *m*/*z* (unknown structure) and 113 *m*/*z* (isobaric compounds: 3-ethyl-cyclopentanone, 2-heptenal, and 1-octene). When setting a lower frying temperature, the emission pattern resembles linear growth.

It is well established that frying temperature affects oil degradation intensity [49], reducing VOC emissions at lower temperatures [29]. At lower temperatures (160 °C), significantly weaker emissions were observed, except for hexanal in palm oil, where similar emission patterns were recorded. Comparable emission curves were noted at both temperatures for the same VOCs, though an anomaly was observed for Cluster 2 in rapeseed oil, where emissions followed a more linear trend at 160 °C.

## 3. Materials and Methods

### 3.1. Materials

The tempeh, a fermented soybean, was obtained/commercially purchased from a local supermarket in Gdansk and manufactured by SoyBean Company in the Netherlands in its frozen form. The tempeh had a moisture content of 55.2 ± 4.2 % (*n* = 8), measured with WPS 30S (Radwag, Radom, Poland). For experimental purposes, the tempeh was cut into a uniform cube of 1.0 × 1.0 ± 0.2 cm and weighed 0.65 ± 0.04 g. The tempeh cubes were stored at −40 °C until usage and fried for 3 min.

Two types of oils were used for the deep-frying analysis, rapeseed oil and palm oil. To ensure consistency throughout the experiment, palm oil from a single large can with an airtight lid was used, while bottles of rapeseed oil from the same manufacturing batch were stored in a dark, cool environment. The quality of the oils was assessed before and after frying by measuring the total polar material (TPM) with a Testo 270 device (Testo, Wien, Austria) and thermal stability based on the Rancimat method (Methrohm, Herisau, Switzerland). The quality parameters of oils are listed in Appendix A.

The chemical standards of ethanol, hexanoic acid, octanoic acid, hexanol, ethyl acetate, acetic acid, 2,3-butadione, 2-ethyl furan, 3-methyl-2-butenal, hexanal, heptanal, 2,5-dimethyl pyrazine, 1-octen-3-one, benzaldehyde, 2-pentyl furan, and trans-2-nonenal were purchased from Sigma Aldrich (Steinheim, Germany).

### 3.2. Experimental Setup

The experimental setup used to measure volatile compounds emitted during frying using a PTR-MS was previously introduced in detail by [34]. In summary, an airtight glass reaction chamber containing a heated glass of frying oil was passed through by water vapour at a constant flow of 0.5 L/min. Tempeh was hung by a single rod to immerse it in the heated oil. The emitted fumes were carried out by the flowing air through a capillary to the PTR-MS. A filter was inserted in between the capillaries to keep the oil droplet and condensate from being transported to the PTR-MS inlet. Different temperatures of 160 and 180 °C were used during the experiment. For the oil reuse, frying was carried out at 180 °C, and the oils were stored at 4 °C between usage without the addition of fresh oil.

### 3.3. PTR-MS

A PTR TOF1000 ultra proton transfer reaction mass spectrometer (Ionicon GmbH, Innsbruck, Austria) was used for the direct analysis of volatile compounds generated during frying. The E/N value of 100 Td was carefully selected and maintained during the experiment. The total flow was set to 1000 sccm, and to facilitate the fume transportation to the PTR system, the clean air dilution flow rate was set to 999 sccm. The transfer line and the drift tube were kept at 70 °C. A full MS spectrum was recorded every second, and the combination of IoniTOF v3.0 software (Ionicon GmbH, Innsbruck, Austria) and PTR-MS Viewer v3.4 (Ionicon GmbH, Innsbruck, Austria) was used to extract the raw data (i.e., corrected cps values).

During data preprocessing, ions higher than 40.00 *m*/*z* were considered for further analysis. Next, the isotope ^13^C ions were removed. Afterwards, standardisation of the data was carried out by subtracting the signals from the one recorded during the first second of food introduction into the oil. This initial signal was used as the zero-second reference point for all subsequent measurements. After that, all experimental runs were combined, producing a dataset with 181 data points totalling 32 identified ions and their matching cps values.

### 3.4. HS-SPME-GC-MS

To comprehensively understand the emitted frying fumes, a system of HS-SPME-GC-MS was adopted. The used oil was kept in a vial for the extraction of the frying fumes with a Carboxen/Polydimethylsiloxane (CAR/PDMS) StableFlex fibre of 85 μm thickness and 2 cm length (Sigma-Aldrich, St. Louis, MO, USA) at 60 °C for a total of 50 min. Afterwards, the extracted volatiles were analysed using an Agilent 7890A (Agilent Technologies, Santa Clara, CA, USA) gas chromatograph coupled to Pegasus 4D TOFMS (LECO Corp., Saint Joseph, MO, USA) and a Gerstel MPS2 auto-sampler (Gerstel, Pforzheim, Germany). Chromatographic separation was achieved using an Equity 1 column (30 m × 0.25 mm × 0.25 μm,) (Supelco Inc., Bellefonte, PA, USA) with a temperature gradient programme of 6 °C/min from 40 to 250 °C. Helium with 99.99% purity (Air Liquide, Kraków, Poland) served as the carrier gas at a constant flow rate of 1.0 mL/min. The transfer line and ion source temperature were set to 250 °C with a detector voltage of 1700 V and an ionisation energy of 70 eV. The spectra were acquired at 10 Hz (equivalent to 1 spectra/s) with an ion mass range of 40–300 *m*/*z*. The raw CDF files were extracted using Leco ChromaTOF 4.51.6.0.

### 3.5. Data Analysis

The PTR-MS and GC-MS raw data were preprocessed using Microsoft Excel 365. Further, the Venn diagram and a heatmap with hierarchical clustering analysis (HCA) were designed using Orange Data Mining version 3.34.0 [56]. Additionally, cosine similarities and circular HCA plots were calculated and generated using R version 3.4.

## 4. Conclusions

The study conclusively demonstrated that the intensity and emission profile of VOCs during deep frying are significantly influenced by variables such as repeated use cycles of the same oil, frying temperature, and oil type. The frying fumes were found to be rich in compounds from the saturated and unsaturated aldehyde groups, including acetaldehyde, acrolein, methacrolein, hexanal, 2,4-heptadienal, 2,4-decadienal, and 2-decenal. This confirms the critical role of fatty acid oxidation in VOC formation during frying. The presence of, e.g., acrolein, raises concerns about the potential adverse health effects from exposure to frying fumes [57], particularly in food preparation environments. However, further studies are needed to quantify the emissions and assess inhalation-related effects under controlled, realistic conditions. Moreover, this study may enhance current research examining global exposure to cooking emissions [58] by providing deeper insights into volatile organic compound (VOC) formation through oil degradation mechanisms specifically under modelled deep-frying conditions.

The complementary use of GC-MS alongside PTR-MS proved effective, allowing simultaneous tracking of the emission profiles of isobaric and isomeric compounds for several recorded ions (i.e., 87 *m*/*z*, 113 *m*/*z*, 115 *m*/*z*, 127 *m*/*z*, 129 *m*/*z*, 141 *m*/*z*, and 143 *m*/*z*). While PTR-MS could benefit from enhanced resolution for isobaric compounds, GC-MS remains invaluable for distinguishing compounds with the same molecular formula.

The algorithm for aggregating ions with similar emission behaviour based on a cosine similarity was successful in identifying fragment ions (e.g., hexanal; Cluster 5), structurally similar compounds (e.g., C_10_-unsaturated aldehydes; Cluster 1), and those with comparable emission profiles. This approach could be applied in other studies using PTR-MS for untargeted real-time monitoring of complex chemical processes.

In general, similar emission patterns were observed for both oils, namely rapeseed and palm oil. However, the consecutive frying and frying temperature seems to be the most impactful on the emission characteristic. Most of the monitored VOCs represent close-to-logarithmic emission, although the few clustered VOCs in Cluster 1 and Cluster 6 exhibit a more linear profile.

In conclusion, the results of this research contribute valuable insights into VOC emissions during deep frying and offer methodological advances that may inspire future studies in food processing and VOC monitoring.

## Figures and Tables

**Figure 1 molecules-29-05046-f001:**
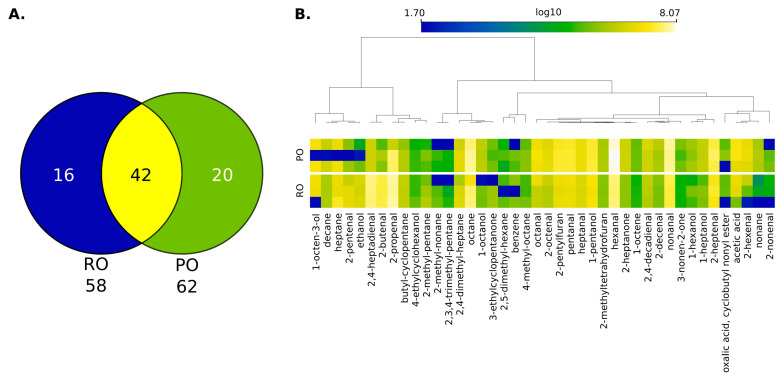
HS-SPME-GC-MS analysis of rapeseed (ROs) and palm oils (POs) used for tempeh frying: (**A**)—Venn diagram of detected VOCs; (**B**)—Heatmap of 42 shared VOCs (features) where colour intensity is log10 transformation of peak areas. Features were clustered using HCA (Euclidean distance, average linkage); n/a values were substituted by noise signal values.

**Figure 2 molecules-29-05046-f002:**
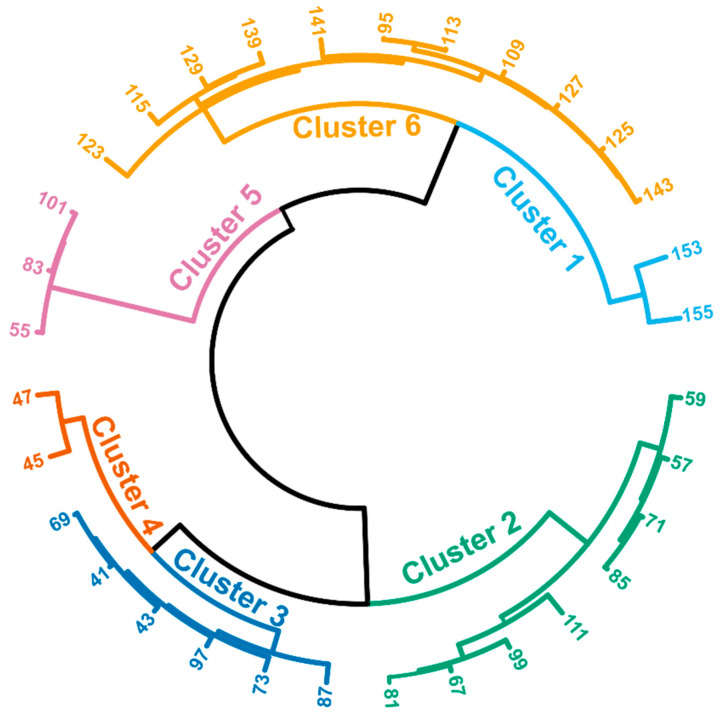
Circular hierarchical cluster analysis (HCA) of the ion signals registered during deep frying of tempeh at various conditions (input matrix size: 6517 × 32; distance: cosine similarity (CS); linkage: Ward; height threshold for cluster cut-off: 25%). The labels describe the nominal *m*/*z* of the registered ions.

**Figure 3 molecules-29-05046-f003:**
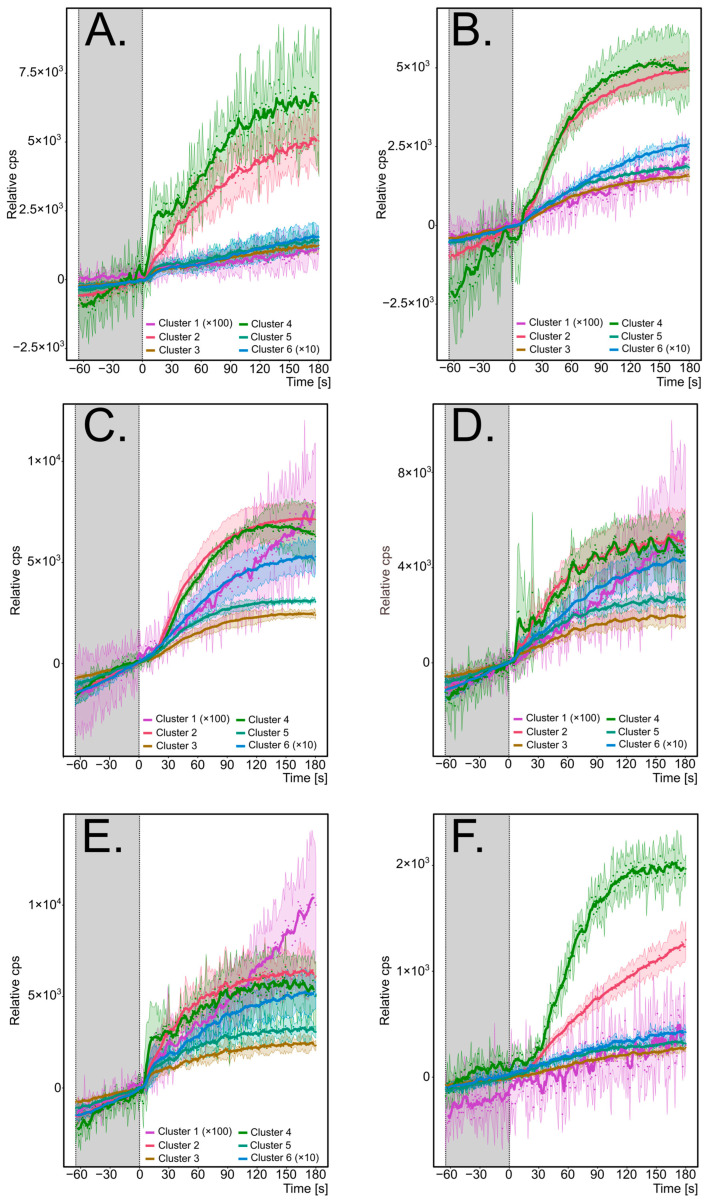
The emission profiles of clustered VOCs monitored during deep frying of tempeh in rapeseed oil: (**A**)—frying at 180 °C, fresh oil; (**B**)—frying at 180 °C, second use of oil; (**C**)—frying at 180 °C, third use of oil; (**D**)—frying at 180 °C, fourth use of oil; (**E**)—frying at 180 °C, fifth use of oil; (**F**)—frying at 160 °C, fresh oil. The signal is a mean value for the volatiles in each cluster. The *y*-axis represents the relative counts per second, where zero is the start of frying; the *x*-axis represents the frying time before frying (marked grey) and after placing tempeh; the line is the result of smoothing with a moving average with a step = 5; and the coloured area represents the standard deviation (*n* = 3). Some signals are multiplied as mentioned in legends.

**Figure 4 molecules-29-05046-f004:**
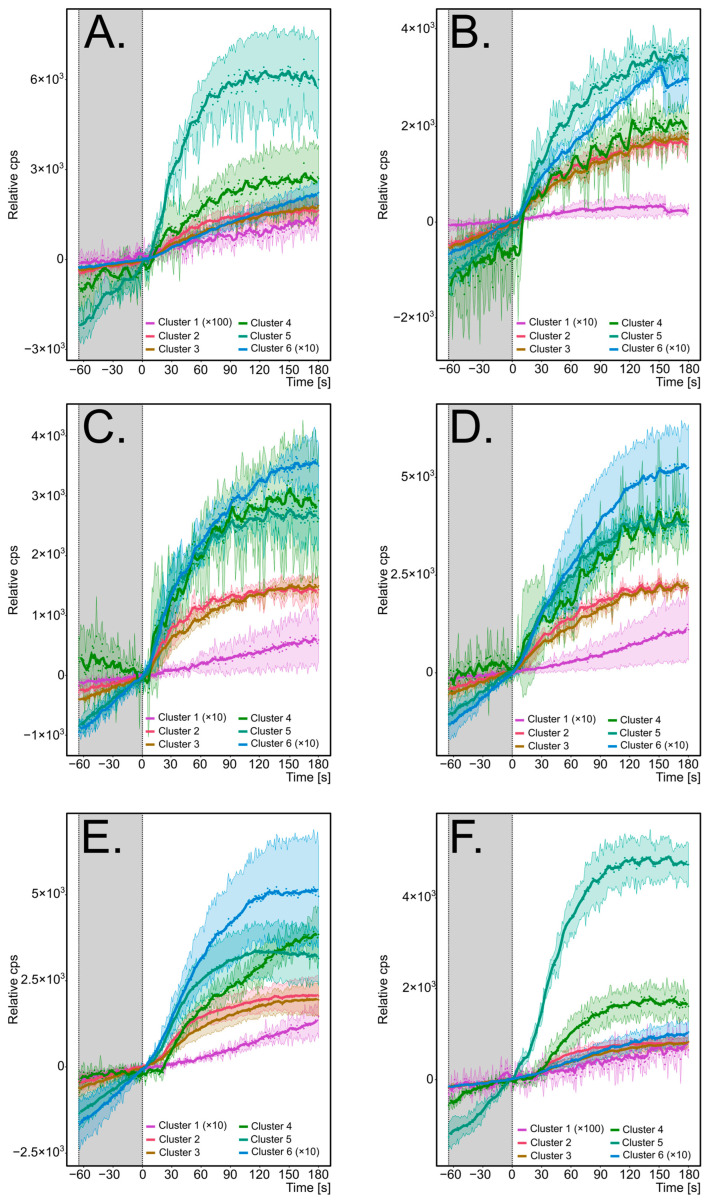
The emission profiles of clustered VOCs monitored during deep frying of tempeh in palm oil: (**A**)—frying at 180 °C, fresh oil; (**B**)—frying at 180 °C, second use of oil; (**C**)—frying at 180 °C, third use of oil; (**D**)—frying at 180 °C, fourth use of oil; (**E**)—frying at 180 °C, fifth use of oil; (**F**)—frying at 160 °C, fresh oil. The signal is a mean value for the volatiles in each cluster. The *y*-axis represents the relative counts per second, where zero is the start of frying; the *x*-axis represents the frying time before frying (marked grey) and after placing tempeh in the oil; the line is the result of smoothing with a moving average with a step = 5; and the coloured area represents the standard deviation (*n* = 3). Some signals are multiplied as mentioned in legends; in (**B**), an unexpected signal drop/rise for some ions is due to the *m*/*z* axis calibration issues.

**Table 1 molecules-29-05046-t001:** Determination of PTR-MS ion signals registered during the monitoring of frying fumes resulted in deep frying of the tempeh; RT is the retention time [min] obtained in the HS-SPME-GC-MS experiment.

Monitored Ion	Cluster No.	Ion *m*/*z*	Molecular Formula	Identification	RT [Min]
41	3	41.04	C_3_H_5_^+^	fragment ion	-
43	3	43.05	C_3_H_7_^+^	fragment ion	-
45	4	45.03	(C_2_H_4_O)H^+^	acetaldehyde ^1^	5.07
47	4	47.05	(C_2_H_6_O)H^+^	ethanol ^2^	5.25
55	5	55.05	C_4_H_7_^+^	hexanal fragment ion water cluster H_7_O_3_^+^ (55.04 *m*/*z*)	-
57	2	57.03	(C_3_H_4_O)H^+^	2-propenal ^2^	5.42
59	2	59.05	(C_3_H_6_O)H^+^	propanal ^3^ [1]	-
67	2	67.05	C_5_H_7_^+^	2,4-heptadienal ion fragment	-
69	3	69.03	(C_4_H_4_O)H^+^	furan ^4^	5.48
71	2	71.05	(C4H6O)H^+^	2-methyl-2-propenal ^4^	6.01
		71.05	(C4H6O)H^+^	2-butenal ^2^	7.03
73	3	73.06	(C_4_H_8_O)H^+^	butanal ^4^	6.23
81	2	81.07	C_6_H_9_^+^	2,4-heptadienal ion fragment	-
83	5	83.09	C_6_H_11_^+^	hexanal fragment ion	-
85	2	85.06	(C_5_H_8_O)H^+^	2-pentenal ^2^	7.74
87	3	87.08	(C5H10O)H^+^	2-methyl tetrahydrofuran ^2^	7.23
		87.08	(C5H10O)H^+^	pentanal ^2^	8.98
		87.12	(C6H14)H^+^	2-methyl-pentane ^2^	5.91
95	6	95.05	(C_6_H_6_O)H^+^	unknown/fragment ion	-
97	3	97.06	(C_6_H_8_O)H^+^	ethyl furan [36]	-
99	2	99.08	(C_6_H_10_O)H^+^	2-hexenal ^2^	11.39
101	5	101.10	(C_6_H_12_O)H^+^	hexanal ^2^	9.96
109	6	109.10	C_9_H_13_^+^	unknown/fragment ion	-
111	2	111.08	(C_7_H_10_O)H^+^	2,4-heptadienal ^2^	15.77
113	6	113.10	(C7H12O)H^+^	3-ethyl-cyclopentanone ^2^	14.41
		113.10	(C7H12O)H^+^	2-heptenal ^2^	14.20
		113.13	(C8H16)H^+^	1-octene ^2^	9.56
115	6	115.11	(C7H14O)H^+^	2-heptanone ^2^	12.22
		115.11	(C7H14O)H^+^	heptanal ^2^	12.59
		115.15	(C8H18)H^+^	octane ^2^	9.74
		115.15	(C8H18)H^+^	2,3-dimethyl-hexane ^2^	8.18
		115.15	(C8H18)H^+^	2,3,4-dimethyl-pentane ^2^	8.68
123	6	123.08	(C_8_H_10_O)H^+^	unknown/fragment ion	-
125	6	125.09	(C_8_H_12_O)H^+^	unknown/fragment ion	-
127	6	127.11	(C8H14O)H^+^	2-octenal ^2^	16.90
		127.15	(C9H18)H^+^	butyl-cyclopentane ^2^	13.44
129	6	129.13	(C8H16O)H^+^	1-octen-3-ol ^2^	14.65
		129.13	(C8H16O)H^+^	octanal ^2^	15.33
		129.13	(C8H16O)H^+^	2,4-dimethyl-heptane ^2^	10.24
		129.16	(C9H20)H^+^	4-methyl-octane ^2^	11.32
		129.16	(C9H20)H^+^	4-ethyl-cyclohexanol ^2^	12.31
		129.16	(C9H20)H^+^	nonane ^2^	15.63
139	6	139.11	(C_9_H_14_O)H^+^	2-pentyl furan ^2^	14.94
141	6	141.13	(C_9_H_16_O)H^+^	2-nonenal ^2^	19.51
		141.13	(C_9_H_16_O)H^+^	3-nonen-2-one ^2^	18.92
143	6	143.14	(C9H18O)H^+^	nonanal ^2^	18.03
		143.18	(C10H22)H^+^	2-methyl-nonane ^2^	14.03
		143.18	(C10H22)H^+^	decane ^2^	15.00
153	1	153.13	(C_10_H_16_O)H^+^	2,4-decadienal ^2^	23.41
155	1	155.14	(C_10_H_18_O)H^+^	2-decenal ^2^	21.99

^1^ determined in rapeseed oil in the HS-SPME-GC-MS experiment; ^2^ determined in both rapeseed and palm oil in the HS-SPME-GC-MS experiment; ^3^ found in the literature but not detected in the HS-SPME-GC-MS experiment; ^4^ determined in palm oil in the HS-SPME-GC-MS experiment.

## Data Availability

The data presented in this study are available upon request from the corresponding author.

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
