# Peer review of "Investigation of the Frying Fume Composition During Deep Frying of Tempeh Using GC-MS and PTR-MS"

_molecules, 2024, doi:10.3390/molecules29215046_

Round 1
Reviewer 1 Report
Comments and Suggestions for Authors
1. The introduction effectively establishes the importance of frying fume analysis, but it could benefit from further clarity. Consider reducing repetitive statements regarding the effects of frying temperature and oil type to enhance readability. For example, the discussion on frying oil degradation could be streamlined to avoid redundant mentions of lipid oxidation.
2. While the manuscript discusses the presence of hazardous VOCs such as acrolein, it would be beneficial to elaborate more on the health implications of these findings. For instance, a section could be dedicated to the potential occupational exposure risks and how these emissions compare to known safety thresholds, particularly in the food industry.
3. The cluster analysis is a strong feature of the manuscript, but further elaboration is needed on the interpretation of the clustered VOCs. The manuscript should provide more specific insights into the chemical pathways leading to these clusters and how they relate to oil degradation processes, which would add depth to the analysis.
Comments on the Quality of English LanguageNo.
Author Response
Comments 1: The introduction effectively establishes the importance of frying fume analysis, but it could benefit from further clarity. Consider reducing repetitive statements regarding the effects of frying temperature and oil type to enhance readability. For example, the discussion on frying oil degradation could be streamlined to avoid redundant mentions of lipid oxidation.
Response 1: Thank you for pointing this out – after carefully examining the text, all of the repetitive statements were removed or modified.
Comments 2: While the manuscript discusses the presence of hazardous VOCs such as acrolein, it would be beneficial to elaborate more on the health implications of these findings. For instance, a section could be dedicated to the potential occupational exposure risks and how these emissions compare to known safety thresholds, particularly in the food industry.
Response 2: We agree with your statement. The additional information in the Results and Discussion section was added.
Comments 3: The cluster analysis is a strong feature of the manuscript, but further elaboration is needed on the interpretation of the clustered VOCs. The manuscript should provide more specific insights into the chemical pathways leading to these clusters and how they relate to oil degradation processes, which would add depth to the analysis.
Response 3: Thank you. More information about the possible pathways was added
Comments 4: The English could be improved to more clearly express the research.
Response 4: We’ve corrected a few language issues in the manuscript. Thank you for your suggestion.
Reviewer 2 Report
Comments and Suggestions for Authors
After reviewing the manuscript “Investigation of the frying fume composition during deep frying of Tempeh using GC-MS and PTR-MS” I have the following comments:
The study presents a relevant investigation into the volatile organic compounds (VOCs) emitted during the deep frying of tempeh, a popular food item with increasing global demand as a meat substitute. The use of proton transfer reaction mass spectrometry (PTR-MS) and gas chromatography-mass spectrometry (GC-MS) together is a strong aspect of the study, offering both real-time and compound-specific data.
Major Issues:
1. Clarity in Data presentation:
Some of the figures, especially those representing the hierarchical clustering analysis (HCA) and emission profiles (Figures 3, and 4), could be improved for clarity. While the clustering is informative, the presentation could benefit from clearer labeling and color coding to make the distinctions between clusters more visually accessible.
2. The Supplementary Figures (S1-S12) provide extensive data on VOC emission profiles; however, the explanations accompanying these figures are limited. Detailed descriptions of key patterns observed in these figures would enhance the reader's ability to interpret the results.
3. The study mentions keeping the oil at the desired temperature for 60 seconds before frying to stabilize the temperature. However, this may not be enough to ensure a uniform temperature distribution in the oil, especially when frying multiple cycles.
4. While the study employs PTR-MS for real-time monitoring, it acknowledges the limitation of PTR-MS in distinguishing isobaric compounds without a separation step. This shortcoming is significant when dealing with complex food matrices where many compounds have similar mass-to-charge ratios.
5. The experimental setup used to fry tempeh and capture VOCs may not accurately reflect real-world cooking conditions, as the process is conducted in a closed system that limits external influences. Conducting follow-up studies in conditions that closely mimic typical cooking environments would make the results more applicable to real-world scenarios.
6. The supplementary material mentions unexpected signal drops for some ions due to m/z axis calibration issues but does not explain how these issues were addressed or their impact on the overall analysis.
7. The manuscript references similar studies using GC-MS and PTR-MS but does not provide a thorough comparison with these studies. It fails to highlight the unique contributions of this research compared to what is already known.
A detailed comparison with relevant literature would help position this study within the broader context of VOC analysis in food processing, emphasizing its unique contributions.
Other Minor issue:
The description of the experimental setup, specifically the placement of the PTR-MS inlet and the control of airflow through the reaction chamber, is somewhat brief. A more detailed explanation would help readers understand the conditions under which the VOCs were analyzed. Including a schematic of the experimental setup and a clearer description of how the airflow was maintained could enhance reproducibility and provide more transparency to the methodology.
Author Response
Comments 1: Clarity in Data presentation:
Some of the figures, especially those representing the hierarchical clustering analysis (HCA) and emission profiles (Figures 3, and 4), could be improved for clarity. While the clustering is informative, the presentation could benefit from clearer labeling and color coding to make the distinctions between clusters more visually accessible.
Response 1: The Figures were modified by changing color coding in Figure 2 (Okabe-Ito color-blind friendly palette), enlarging fonts (all Figures) and increasing the line width (Figures 3 and 4).
Comments 2: The Supplementary Figures (S1-S12) provide extensive data on VOC emission profiles; however, the explanations accompanying these figures are limited. Detailed descriptions of key patterns observed in these figures would enhance the reader's ability to interpret the results.
Response 2: The detailed explanation of the emission pattern was done in the body text but for the average clustered signal previously. To address your comment more detailed description with providing worth-mentioning comments on particular VOCs was placed in Results and Discussion.
Comments 3: The study mentions keeping the oil at the desired temperature for 60 seconds before frying to stabilize the temperature. However, this may not be enough to ensure a uniform temperature distribution in the oil, especially when frying multiple cycles.
Response 3: Thank you for such a detailed examination of our paper. The measuring probe (K-type thermocouple) was placed inside the oil bulk (and not touching the surface of a beaker). Thus, the temperature was measured within the oil and the PID-regulated heater provided a stabilized temperature for the whole experiment. This was in detail explained (together with the figure) in our paper entitled: Release Kinetics Studies of Early-Stage Volatile Secondary Oxidation Products of Rapeseed Oil Emitted during the Deep-Frying Process (doi:10.3390/molecules2604100). Since it is an open-access article we’ve decided to leave the description in the current shape.
Comments 4: While the study employs PTR-MS for real-time monitoring, it acknowledges the limitation of PTR-MS in distinguishing isobaric compounds without a separation step. This shortcoming is significant when dealing with complex food matrices where many compounds have similar mass-to-charge ratios.
Response 4: We strongly agree with this. That is why the coupling GC-MS with the PTR-MS experiment provides more accurate measurements. As we mentioned in Table 1 one ion signal can describe more than one compound. We are aware of that. We’ve mentioned this issue in the introduction: The absence of a specific separation step within PTR-MS and its reliance on distinguishing compounds solely through their mass-to-charge (m/z) values often underestimate the volatile composition. Thus, combining PTR-MS with GC-MS for enhanced accuracy and compound identification is advisable.
And in Conclusions: The complementary use of GC-MS alongside PTR-MS proved effective, allowing simultaneous tracking of the emission profiles of isobaric and isomeric compounds for several recorded ions (i.e. 87 m/z, 113 m/z, 115 m/z, 127 m/z, 129 m/z, 141 m/z, 143 m/z). While PTR-MS could benefit from enhanced resolution for isobaric compounds, GC-MS remains invaluable for distinguishing compounds with the same molecular formula.
However, we are ready for clarification if there is a need for that.
Comments 5: The experimental setup used to fry tempeh and capture VOCs may not accurately reflect real-world cooking conditions, as the process is conducted in a closed system that limits external influences. Conducting follow-up studies in conditions that closely mimic typical cooking environments would make the results more applicable to real-world scenarios.
Response 5: We are aware that the real conditions aren’t comparable to the proposed ones. However, since there is a scientific gap in this field, it is needed to focus on simplified measuring conditions – limiting the variables (room size, ventilation, other activities) and focusing on the true process. You can find the experiment in the mimic conditions such as a recent paper Exposure to cooking emitted volatile organic compounds with recirculating and extracting ventilation solutions (doi.org/10.1016/j.buildenv.2024.111743). We’ve decided to cite this paper in the Conclusions together with the appropriate passage.
Comments 6: The supplementary material mentions unexpected signal drops for some ions due to m/z axis calibration issues but does not explain how these issues were addressed or their impact on the overall analysis.
Response 6: We’ve carefully examined the influence of this drop and decided that it is not affecting the clustering of the data. Since this drop appears only for some of the ions independently on the cluster, we can state that wasn’t a key factor for clustering. However, we’ve decided not to focus on the last seconds of frying in this situation, but at the same time leave the true data as they are. Should we provide more details on that?
Comments 7 and 8: The manuscript references similar studies using GC-MS and PTR-MS but does not provide a thorough comparison with these studies. It fails to highlight the unique contributions of this research compared to what is already known.
A detailed comparison with relevant literature would help position this study within the broader context of VOC analysis in food processing, emphasizing its unique contributions.
Response 7 and 8: Thank you. We’ve decided to add the broader context and point out the novelty of our studies.
Comments 9: The description of the experimental setup, specifically the placement of the PTR-MS inlet and the control of airflow through the reaction chamber, is somewhat brief. A more detailed explanation would help readers understand the conditions under which the VOCs were analyzed. Including a schematic of the experimental setup and a clearer description of how the airflow was maintained could enhance reproducibility and provide more transparency to the methodology.
Response 9: We agree. However, as we stated already the setup was described in the previous paper in Molecules (doi:10.3390/molecules2604100), and since it’s open access we don’t want to double the information. Please let us know if it is fine for you.
Round 2
Reviewer 1 Report
Comments and Suggestions for Authors
After evaluation, the authors have addressed the issues raised by the reviewers and the manuscript is acceptable.
Author Response
Thank you for your expertise and great involvement in revising our manuscript. We are grateful!
Reviewer 2 Report
Comments and Suggestions for Authors
After reviewing the revised version of the manuscript and supplementary information, I have the following comments:
1. The authors provided additional explanations in response to comments, including detailing the setup and referencing prior studies. The real-world applicability of the results remains a limitation, but the authors acknowledge this and have suggested future work to address it.
2. Although the figures and data presentation were improved, the graphs, particularly those showing clustered emission profiles, remain somewhat cluttered and hard to interpret. For figures 3 and 4, only a minor line width was changed. However, the text inside the figures are still unclear. Also, the axis legends need improvements.
Author Response
Comments 1. The authors provided additional explanations in response to comments, including detailing the setup and referencing prior studies. The real-world applicability of the results remains a limitation, but the authors acknowledge this and have suggested future work to address it.
Response 1: Indeed, this is the main limitation of our work. Thank you for addressing it. We hope the acknowledged work is enough for you and we can close this issue.
Comments 2: Although the figures and data presentation were improved, the graphs, particularly those showing clustered emission profiles, remain somewhat cluttered and hard to interpret. For figures 3 and 4, only a minor line width was changed. However, the text inside the figures are still unclear. Also, the axis legends need improvements.
Response 2: After in-depth validation, we've provided modifications to Figure 3 and Figure 4 as follows:
1. Increase the font size for the axis and legend
2. Add a dotted line for the 'grey' area
3. Add a line for the 'SD area'
4. Switching to 2 (rows) x 3 (verses) to fully cover one page of the manuscript
We hope that this enhances the clearness of those figures.